# Van der Waals isotope heterostructures for engineering phonon polariton dispersions

M. Chen [1,8], Y. Zhong[2,8], E. Harris[3], J. Li [4], Z. Zheng[5], H. Chen [2,6], J.-S. Wu[7], P. Jarillo-Herrero [5], Q. Ma [3], J. H. Edgar [4], X. Lin [2] & S. Dai [1] ✉

Element isotopes are characterized by distinct atomic masses and nuclear spins, which can significantly influence material properties. Notably, however, isotopes in natural materials are homogenously distributed in space. Here, we propose a method to configure material properties by repositioning isotopes in engineered van der Waals (vdW) isotopic heterostructures. We showcase the properties of hexagonal boron nitride (hBN) isotopic heterostructures in engineering confined photon-lattice waves—hyperbolic phonon polaritons. By varying the composition, stacking order, and thicknesses of $h^{10}BN$ and $h^{11}BN$ building blocks, hyperbolic phonon polaritons can be engineered into a variety of energy-momentum dispersions. These confined and tailored polaritons are promising for various nanophotonic and thermal functionalities. Due to the universality and importance of isotopes, our vdW isotope heterostructuring method can be applied to engineer the properties of a broad range of materials.

Element isotopes exist universally and distribute evenly in space (Fig. 1a). Various isotopes have the same number of protons (and electrons) but differ in the numbers of neutrons in the nuclei. Therefore, they possess distinct atomic masses and nuclear spins. The variation in atomic mass affects the nuclear structures[1,2], rate of chemical reactions[3], and lattice vibrations[4–6] that govern thermal[7–10] and elastic responses[11]. Electronic properties[12], including superconductivity[13], transport[14], excitons[10,15], and band structures, also vary with isotopes via electron-phonon interactions. In addition, isotopes offer an important resource for quantum technologies[16–20] by altering the nuclear spin-related properties for quantum preservation and qubit encoding. These isotope-related merits were facilitated by isotope purifications that globally alter the overall isotope ratios in materials[1–20]. Notably, both naturally abundant and isotopically purified crystals are isotope-homogenous: all isotopes distribute evenly in space.

In this work, we explore the isotope spatial-heterogeneity by establishing a materials engineering method called van der Waals (vdW) isotope heterostructuring. vdW isotope heterostructuring configures material properties by repositioning isotopes in engineered isotopic heterostructures (Fig. 1b). The advance of this method lies in that fundamental material parameters—atomic mass and nuclear spin— of the same element can vary internally in materials and be spatially engineered. Therefore, intrinsic material responses can be reshaped from an internal perspective to offer virtues absent in current isotope-homogenous systems. We showcase vdW isotope heterostructuring in engineering confined photon-lattice waves—hyperbolic phonon polaritons (HPPs)[21–30]—in hexagonal boron nitride (hBN) isotopic heterostructures. By varying the composition, stacking, and thicknesses of monoisotopic $^{10}B$ and $^{11}B$ vdW building blocks, HPPs can be engineered into a variety of energy ($\omega$)-momentum ($k$) dispersions that break the universal even-dispersion in hyperbolic materials[31–40]. These

[1]Materials Research and Education Center, Department of Mechanical Engineering, Auburn University, Auburn, AL 36849, USA. [2]Interdisciplinary Center for Quantum Information, State Key Laboratory of Modern Optical Instrumentation, ZJU-Hangzhou Global Science and Technology Innovation Center, Zhejiang University, Hangzhou 310027, China. [3]Department of Physics, Boston College, Chestnut Hill, Massachusetts, MA 02467, USA. [4]Tim Taylor Department of Chemical Engineering, Kansas State University, Manhattan, KS 66506, USA. [5]Department of Physics, Massachusetts Institute of Technology, Cambridge, Massachusetts, MA 02139, USA. [6]International Joint Innovation Center, The Electromagnetics Academy at Zhejiang University, Zhejiang University, Haining 314400, China. [7]Department of Photonics and Institute of Electro-Optical Engineering, National Yang Ming Chiao Tung University, Hsinchu 30050, Taiwan. [8]These authors contributed equally: M. Chen, Y. Zhong. ✉e-mail: sdai@auburn.edu

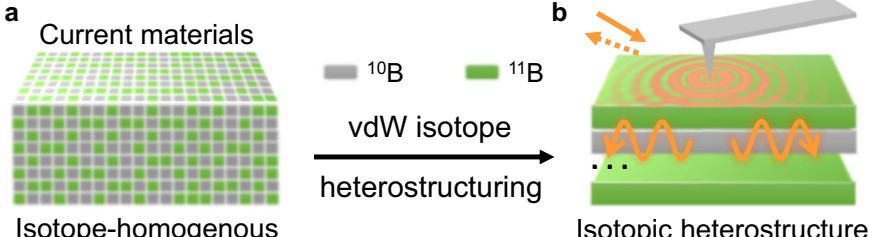

**Fig. 1 | The schematic of van der Waals isotope heterostructuring showcased in phonon polaritons in hBN isotopic heterostructures. a** current materials are isotope-homogenous: isotopes always distribute evenly in space. **b** experiment schematic of s-SNOM imaging of engineered hyperbolic phonon polaritons in $^{10}$B–$^{11}$B isotopic heterostructures.

confined and tailored polaritons may offer on-demand nano-light for various nanophotonic and thermal functionalities. The method of vdW isotope heterostructuring showcased here can also apply to a broad range of materials and properties due to the universality and importance of isotopes.

## Results

### Imaging of phonon polaritons in isotopic heterostructures

Engineered HPPs in isotopic heterostructures were characterized by infrared nano-imaging using scattering-type scanning near-field optical microscopy (s-SNOM, method). The s-SNOM is an illuminated atomic force microscope (AFM) that simultaneously delivers topography and nano-optical image of the underneath sample (Fig. 1b). In the experiment, the AFM tip acts as an antenna[41] to bridge the momentum mismatch and transfer energy between free-space infrared (IR) light (wavelength $\lambda_0$ and frequency $\omega = 1/\lambda_0$) and HPPs[42,43]. The s-SNOM observable near-field amplitude $S(\omega)$ possesses a spatial resolution of ~10 nm. Therefore, it can map HPPs and other nano-optical phenomena in real space.

The crystalline anisotropy of hBN leads to natural hyperbolicity ($\varepsilon_t \varepsilon_z < 0$, $\varepsilon_t = \varepsilon_x = \varepsilon_y$ and $\varepsilon_z$ are in-plane and vertical permittivity, respectively) and HPPs inside the Restrahlen bands[21,25]. HPPs are typically imaged by s-SNOM as parallel fringes[21-25]—standing wave interference between tip-launched and edge-reflected HPPs (Fig. 1b). The fringe period is proportional to the HPP wavelength $\lambda_p$. In the s-SNOM images (Fig. 2), fringes show the strongest oscillation closest to crystal edges, followed by weakly damped ones away from the edges. These characteristics are revealed evidently in s-SNOM profiles (gray and green curves in Fig. 2c, d)—line cuts of the s-SNOM images (gray and blue dashed lines in Fig. 2a, b)—of the $^{10}$B and $^{11}$B hBN. HPPs in $^{10}$B and $^{11}$B hBN exhibit different energy ($\omega$)-momentum ($k = 2\pi/\lambda_p$) dispersions: they show different $\lambda_p$ at $\omega = 1440$ cm$^{-1}$ and 1420 cm$^{-1}$ (Fig. 2a, b) and HPPs at $\omega = 1380$ cm$^{-1}$ and 1385 cm$^{-1}$ can only be imaged in $^{11}$B hBN (Fig. 2c, d). This dispersion variation originates from the different atomic masses that yield Restrahlen band $\omega = 1394.5$ cm$^{-1}$ to 1650 cm$^{-1}$ and 1359.8 cm$^{-1}$ to 1608.7 cm$^{-1}$, for $^{10}$B and $^{11}$B hBN, respectively[44].

The dispersion variation between $^{10}$B and $^{11}$B hBN facilitates the engineering of HPPs in their stacked hybrids—isotopic heterostructures. At representative IR frequencies (Fig. 2), HPPs were imaged in $^{10}$B–$^{11}$B isotopic heterostructures with distinctive characteristics. Close to the edges of isotopic heterostructures (e.g., Fig. 2a, b), HPPs exhibit short-period beats in addition to long-period fringes. As detailed in the s-SNOM profiles (Fig. 2c, d), the short-period beats in $^{10}$B–$^{11}$B isotopic heterostructures (black, red, blue, cyan, and pink curves) show narrower oscillation features than the fringes in single slab $^{10}$B and $^{11}$B hBN (gray and green curves). These beats typically do not appear in s-SNOM images of hBN[21-24]. Moreover, HPPs in isotopic heterostructures depend strongly on the composition and stacking of the $^{10}$B and $^{11}$B building blocks. HPPs in the 2-slab heterostructure with stacking (from top to bottom) $^{10}$B | $^{10}$B, $^{10}$B | $^{11}$B, $^{11}$B | $^{10}$B, $^{11}$B | $^{11}$B, 3-slab

heterostructure $^{10}$B | $^{11}$B | $^{11}$B,   $^{11}$B | $^{10}$B | $^{11}$B,   $^{11}$B | $^{11}$B | $^{10}$B,   $^{10}$B | $^{10}$B | $^{11}$B, $^{10}$B | $^{11}$B | $^{10}$B, $^{11}$B | $^{10}$B | $^{10}$B, and 4-slab heterostructure $^{11}$B | $^{10}$B | $^{11}$B | $^{10}$B and $^{11}$B | $^{10}$B | $^{11}$B | $^{10}$B, show unique s-SNOM oscillations (Fig. 2) that all differ from each other.

### Engineer polariton dispersions in isotopic heterostructures

vdW isotope heterostructuring of HPPs is unambiguously revealed in the $\omega$-$k$ dispersions of isotopic heterostructures that are unique at each composition and stacking (Fig. 3). The experimental data (red circles) by Fourier Transform of the s-SNOM images and profiles (Supplementary Note 1) agree excellently with the electromagnetics (EM) calculation of the complex reflectivity of the isotopic heterostructures (false color, Supplementary Note 2). HPPs can be engineered by varying the composition of the isotopic heterostructures: 1 $^{10}$B slab + 1 $^{11}$B slab (Fig. 3a, b), 1 $^{10}$B slab + 2 $^{11}$B slabs (Fig. 3c–e), 2 $^{10}$B slab + 1 $^{11}$B slabs (Fig. 3f–h), and 2 $^{10}$B slab + 2 $^{11}$B slabs (Fig. 3i, j), all reveal unique dispersions.

Note that current hyperbolic systems have been following the universal even $\omega$-$k$ dispersion[33] (Fig. 4): adjacent polariton branches are separated by identical momentum $\Delta k$ (red arrows). This universal response originates from the Fabry-Perot quantization condition[21]:

$$\Delta\varphi = 2k_z d + \varphi_{R\_sub} + \varphi_{R-\sup} = 2\pi N. \tag{1}$$

$\Delta\varphi$ is the phase accumulation of HPPs, $k_z = \sqrt{\varepsilon_t(\omega/c)^2 - k^2 \varepsilon_t/\varepsilon_z}$ is the vertical momentum, $d$ is the thickness of the hyperbolic slab, $\varphi_{R\_sub}$ and $\varphi_{R\_sub}$ are the phase of HPP reflection at the substrate and superstrate, and $N = 1$, 0, and −1, etc., is an integer and varies at different dispersion branches (e.g., Figs. 3–4). At large $k$, $k_z \cong ik\sqrt{\varepsilon_t/\varepsilon_z}$ and $k = \frac{i}{d}\sqrt{\frac{\varepsilon_z}{\varepsilon_t}}(-\pi N + \frac{\varphi_{R\_sub} + \varphi_{R-\sup}}{2})$. Therefore, adjacent HPP branches are evenly separated by an identical momentum $\frac{i}{d}\sqrt{\frac{\varepsilon_z}{\varepsilon_t}}\pi$—the universal even-dispersion. Although HPPs can be altered by state-of-the-art methods of substrate engineering[45,46], coupling with graphene[47] and phase change materials[48-50], HPP branches therein were altered to the same degree: they are shifted by an identical $\Delta k$ and still follow the universal even-dispersion.

Remarkably, vdW isotope heterostructuring introduced here breaks this universal response by providing HPPs with a variety of dispersions. In $\omega = 1350-1400$ cm$^{-1}$, $^{10}$B | $^{10}$B | $^{11}$B (Fig. 3f) shows only 1 polariton branch instead of the conventional unlimited numbers of branches in hyperbolic materials. $^{10}$B | $^{11}$B | $^{11}$B (Fig. 3c) and $^{10}$B | $^{11}$B | $^{10}$B | $^{11}$B (Fig. 3j) show 2 branches with stronger $N = 1$ branches. $^{11}$B | $^{10}$B | $^{11}$B (Fig. 3d) exhibits approaching branches (see black arrows as a guide to the eye): the $N = 1$ and $N = 0$ branches are merging where the $N = 1$ branch fades at 1390 cm$^{-1}$ while the $N = 0$ one extends to the higher $\omega$. $^{10}$B | $^{11}$B | $^{10}$B | $^{11}$B in Fig. 5e and other compositions in Supplementary Note 3 reveal merging branches: adjacent polariton branches merges. In addition, $^{11}$B | $^{10}$B | $^{11}$B | $^{10}$B (Fig. 3i) supports unbalanced

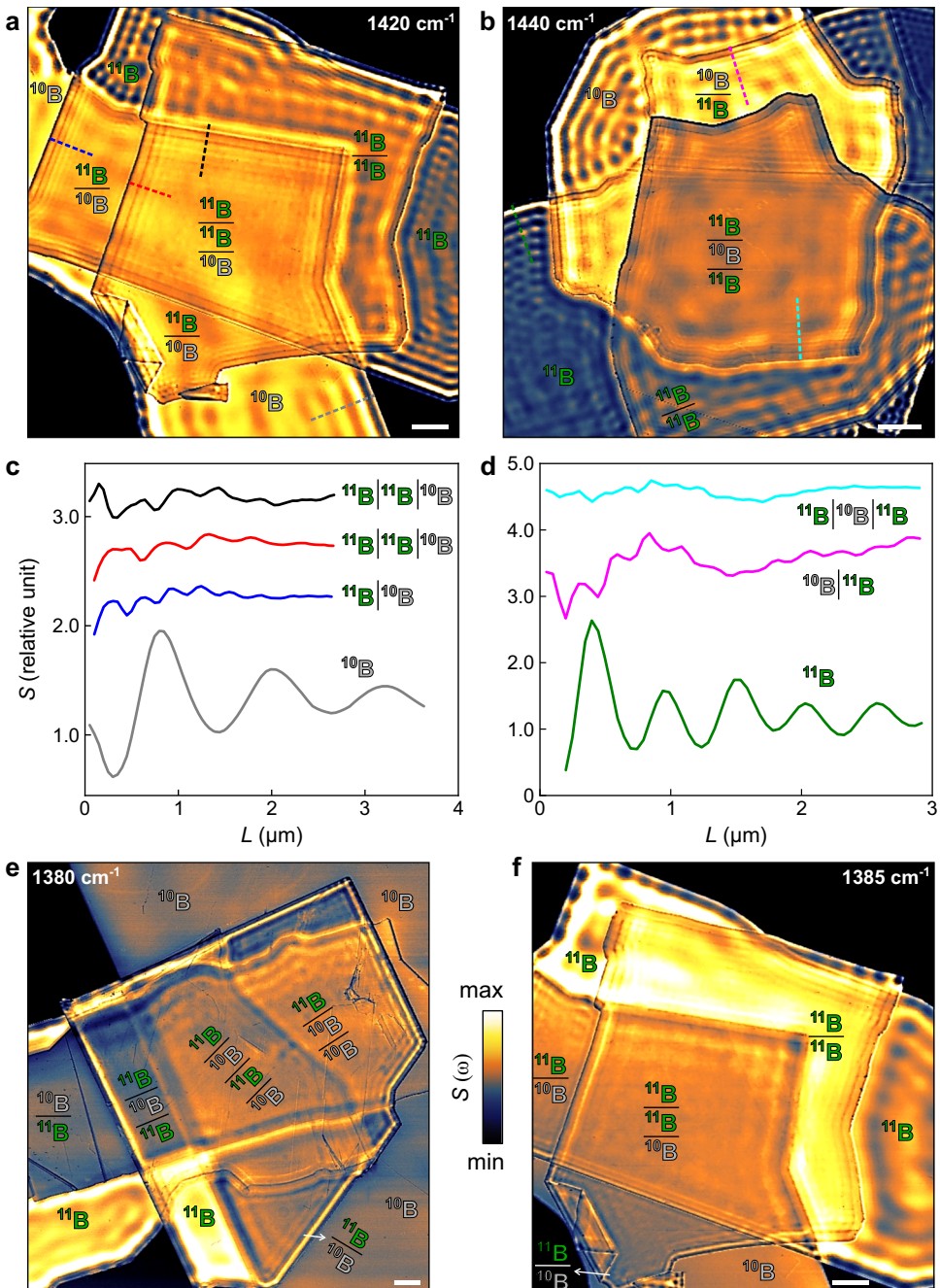

**Fig. 2 | The scattering-type scanning near-field optical microscopy (s-SNOM) nano-infrared images of phonon polaritons in hBN isotopic heterostructures.** s-SNOM amplitude images of $^{10}$B–$^{11}$B isotopic heterostructures at representative infrared (IR) frequency ω = 1440 cm$^{-1}$ (**a**), 1420 cm$^{-1}$ (**b**), 1380 cm$^{-1}$ (**e**), and 1385 cm$^{-1}$ (**f**). **c**, **d** s-SNOM profiles cut along dashed lines in (**a**) and (**b**) reveal short-period beats in $^{10}$B–$^{11}$B isotopic heterostructures (black, red, blue, cyan, and pink) and long-period fringes in single slab $^{10}$B and $^{11}$B hBN (gray and green). *L* is the distance from the slab or heterostructure edge. Scale bar: 2 μm. The compositions of $^{10}$B and $^{11}$B building blocks in each region are denoted by the stacking from top to bottom.

branches: the *N* = 0 branch is significantly weaker than the other polariton branches.

The advanced capabilities of vdW isotope heterostructuring in engineering polariton ω-*k* dispersions originate from the fundamental virtue that atomic masses of the same element inside polaritonic systems can vary locally and be spatially engineered. Specifically, the atomic mass spatial-heterogeneity modifies the local permittivity, thus locally varying $k_z$ in $^{10}$B and $^{11}$B subregions and introducing additional phase jumps of HPP transmission and reflection ($φ_{T\_10/11}$ and $φ_{R\_10/11}$) at $^{10}$B–$^{11}$B interfaces inside the isotopic heterostructures (see Supplementary Note 4 for the detailed analysis). Therefore, the original Fabry-

Perot quantization condition (Eq. 1) is modified. HPPs in $^{10}$B–$^{11}$B isotopic heterostructures can differ from those in isotope-homogenous systems and be engineered into a variety of ω-*k* dispersions that break the universal even-dispersion.

In addition to the composition, HPPs can be engineered by varying the stacking of $^{10}$B and $^{11}$B building blocks. For 2-slab heterostructures with the same composition of 1 $^{10}$B slab + 1 $^{11}$B slab (Fig. 3a, b), $^{11}$B | $^{10}$B stacking (Fig. 3a) supports multiple HPP branches spanning from ω = 1360 to 1400 cm$^{-1}$, whereas $^{10}$B | $^{11}$B stacking (Fig. 3b) supports a single branch at ω = 1360 to 1380 cm$^{-1}$. In 3-slab heterostructures with the composition of 1 $^{10}$B slab + 2 $^{11}$B slabs, three

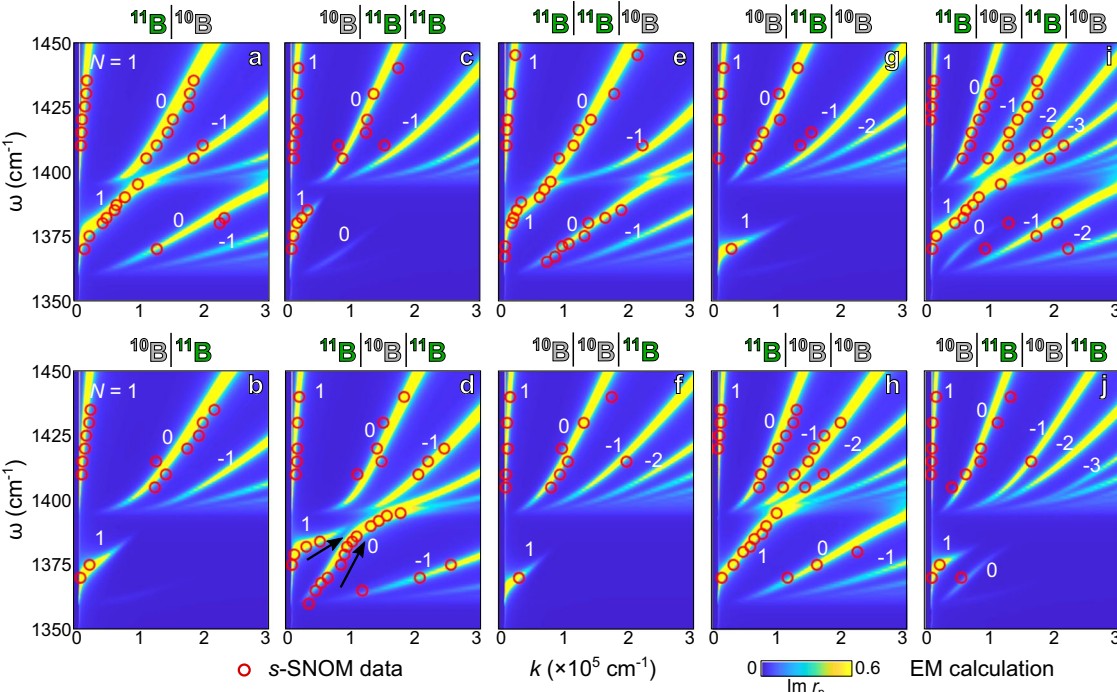

**Fig. 3 | Energy-momentum (ω-k) dispersions reveal the tailoring of phonon polaritons in isotopic heterostructures by varying the compositions and stacking of the $^{10}$B and $^{11}$B building blocks.** Engineered phonon polaritons in 2-slab (**a**, **b**), 3-slab (**c–h**), and 4-slab (**i**, **j**) isotopic heterostructures with the composition and stacking denoted from top to bottom (above the dispersion plots). Various polariton branches are marked by $N = 1$, 0, and −1, etc. Black arrows in (**d**) indicate the two dispersion branches approach together. The s-SNOM experimental data and modeling results from the electromagnetics (EM) calculation of the reflectivity are plotted with red circles and false-color maps, respectively.

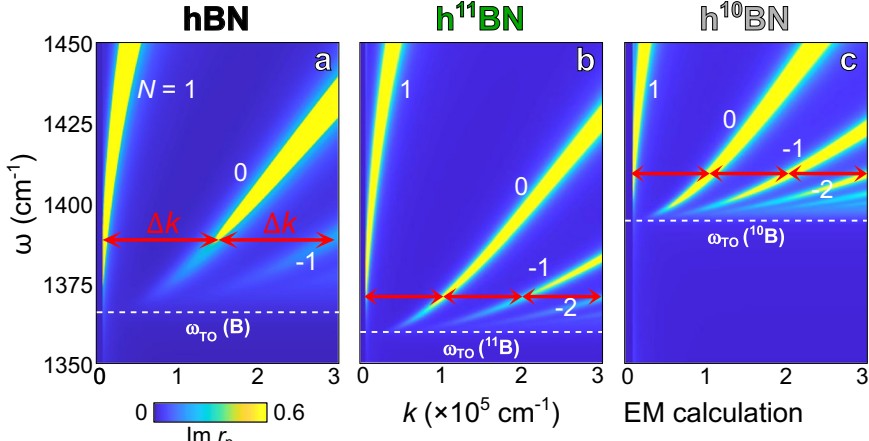

**Fig. 4 | Universal even energy-momentum (ω-k) dispersions in isotope-homogenous hyperbolic systems. a–c** The ω-k dispersions of naturally abundant hBN, h$^{11}$BN, and h$^{10}$BN, respectively. Thickness: 50 nm. Adjacent polariton branches are separated evenly by identical momentum $\Delta k$ (red arrows) in all panels. White dashed lines indicate the transverse optical (TO) phonon frequencies of naturally abundant hBN, h$^{10}$BN, and h$^{11}$BN, respectively. Various polariton branches are marked by $N = 1$, 0, and −1, etc.

types of stackings possess all different dispersions: 2 branches for $^{10}$B | $^{11}$B | $^{11}$B (Fig. 3c), approaching branches for $^{11}$B | $^{10}$B | $^{11}$B (Fig. 3d), and evenly distributed branches for $^{11}$B | $^{11}$B | $^{10}$B (Fig. 3e). This stacking engineering is also revealed in 3-slab heterostructures of 2 $^{10}$B slabs + 1 $^{11}$B slab (Fig. 3f–h) and 4-slab heterostructures of 2 $^{10}$B slabs + 2 $^{11}$B slabs (Fig. 3i, j): each stacking leads to unique HPP dispersion.

vdW isotope heterostructuring is not limited to varying the composition and stacking. The thickness of each building block is another degree of freedom to engineer the overall properties of the isotopic heterostructure. In Fig. 5a-c, we plot the combined experimental and theoretical dispersions in 3-slab isotopic heterostructures with identical composition and stacking $^{11}$B | $^{10}$B | $^{11}$B yet different thicknesses of the $^{10}$B and $^{11}$B building blocks. These dispersions show evident thickness dependence. The thickness of 27 | 23 | 35 nm (for $^{11}$B | $^{10}$B | $^{11}$B, Fig. 5a) exhibits approaching branches: the $N = 1$ branch stops at ω ~ 1385 cm$^{-1}$, whereas the $N = 0$ branch extends into ω > 1400 cm$^{-1}$. The thickness of 25 | 27 | 16 nm (Fig. 5b) exhibits unbalanced branches: the $N = 1$ branch extends into ω > 1400 cm$^{-1}$, whereas the $N = 0$ branch is relatively weak and stops at ω ~ 1380 cm$^{-1}$. Another thickness of 6 | 11 | 9 nm (Fig. 5c) shows

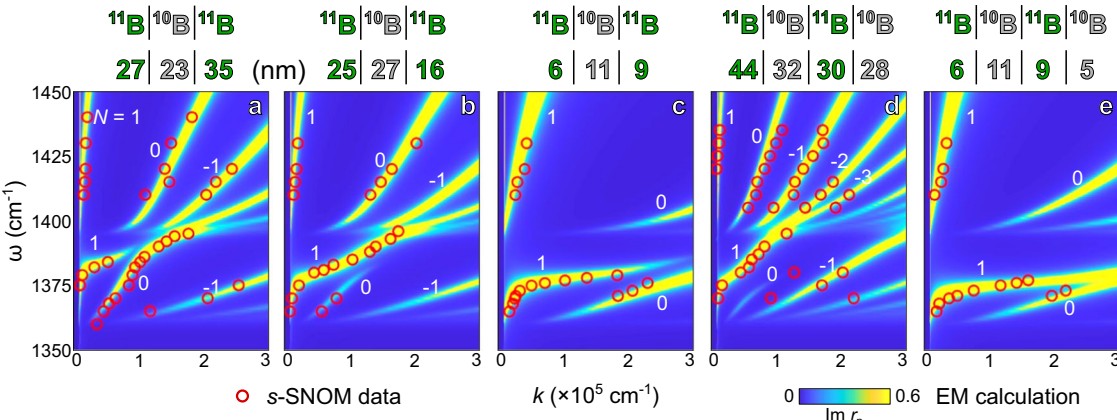

**Fig. 5 | Energy-momentum (ω-*k*) dispersions reveal the tailoring of phonon polaritons in isotopic heterostructures by varying the thicknesses of the ¹⁰B and ¹¹B building blocks. a–d** Phonon polaritons in 3-slab isotopic heterostructures with the identical composition and stacking ¹¹B | ¹⁰B | ¹¹B but with the thickness of 27 | 23 | 35 nm (**a**), 25 | 27 | 16 nm (**b**), and 6 | 11 | 9 nm (**c**). **d**, **e** Phonon polaritons in 4-slab isotopic heterostructures with the identical composition and stacking ¹¹B | ¹⁰B | ¹¹B | ¹⁰B but with the thickness of 44 | 32 | 30 | 28 nm (**d**) and 6 | 11 | 9 | 5 nm (**e**). Various polariton branches are marked by *N* = 1, 0, and −1, etc. The s-SNOM data and modeling results are plotted with red circles and false-color maps, respectively.

balanced and approaching branches, and all HPP branches locate at larger *k*. The degree of freedom of building blocks' thicknesses is further verified in 4-slab isotopic heterostructures ¹¹B | ¹⁰B | ¹¹B | ¹⁰B: the thickness of 44 | 32 | 30 | 28 nm (Fig. 5d) shows unbalanced branches, whereas the thickness of 6 | 11 | 9 | 5 nm (Fig. 5e) shows balanced and merging branches.

## Discussion

Combined experiments and theory in Figs. 1–5 showcase the method of vdW isotope heterostructuring in engineering HPPs in ¹⁰B-¹¹B isotopic heterostructures. A variety of energy-momentum dispersions are engineered to break the universal even-dispersion for hyperbolic systems and those engineered by state-of-the-art methods[45–49]. These unique HPP dispersions engineered in isotopic heterostructures can be attributed to the EM interactions of the ¹⁰B and ¹¹B hBN building blocks. As a result, the standard hyperbolic modes in ¹⁰B and ¹¹B building blocks repulse and form ω-*k* dispersions. Importantly, the delicate EM interactions and mode repulsion depend strongly on the composition, stacking, and thickness of the ¹⁰B and ¹¹B building blocks, therefore offering the degrees of freedom to tailor HPPs in ¹⁰B-¹¹B isotopic heterostructures. In Supplementary Note 5, we provide a detailed analysis of various ω-*k* dispersions of HPPs in ¹⁰B-¹¹B isotopic heterostructures from the perspective of EM interactions.

The highly-confined HPPs may be delicately engineered to offer on-demand nano-light by programming the isotope building blocks in isotopic heterostructures. Therefore, the HPPs can possess unique and tailorable energy-momentum dispersions and photonic density of states. These merits offer unique approaches to control nanoscale light-matter wave propagation, light emission, quantum optics, and energy transfer, thus expanding the current nanophotonic and thermal functionalities[51]. Due to the universality and importance of isotopes, the method of vdW isotope heterostructuring established here can apply to a broad range of materials and engineer thermal, electronic, magnetic, quantum, and other properties where atomic mass or nuclear spin plays a role. Future works may be directed towards extending vdW isotope heterostructuring from hBN to other materials where a series of monoisotopic crystals[4,7–10,15] have been produced. Yet, their isotopic heterostructures are promising but remain unexplored. It is also worth building few-atomic-layer isotopic heterostructures where isotopic moiré superlattices can be involved to configure related material properties, including moiré superconductivity[52], ferroelectricity[53], ferromagnetism[54], excitons[55–58], and many others[59–64]. In addition, it may be valuable to investigate the optical responses of

interfaces within isotopic heterostructures, particularly at frequencies where the permittivity of the constituent slabs changes sign. Furthermore, while this work mainly exploits spatially engineered atomic masses to showcase vdW isotope heterostructuring, the other fundamental virtue of this method−spatially engineered nuclear spins−is equally promising and can be explored in spintronics[18,65–67], chemical reactions[3,68], and quantum information and technologies[16–20].

## Methods
### Fabrication of isotopic heterostructures
¹⁰B-¹¹B isotopic heterostructures were assembled using the standard vdW dry transfer method[52]. First, thin slabs of h¹⁰BN and h¹¹BN were mechanically exfoliated from bulk monoisotopic crystals grown at atmospheric pressure from an iron-chromium flux[69]. Poly (bisphenol A carbonate) (PC)/polydimethylsiloxane (PDMS) were then used to pick up h¹⁰BN and h¹¹BN thin slabs and stack them together into ¹⁰B-¹¹B isotopic heterostructures.

### Infrared nano-imaging
The infrared nano-imaging of polaritons in isotopic heterostructures was performed using the scattering-type scanning near-field optical microscope (s-SNOM, www.neaspec.com). The s-SNOM is a tapping-mode atomic force microscope (AFM) illuminated by monochromatic mid-IR quantum cascade lasers (QCLs) with a frequency coverage of 900−2300 cm⁻¹. In the experiment, the AFM tip (radius ~10 nm, PtIr coating) taps at the frequency of ~280 kHz and the amplitude of ~70 nm. The s-SNOM nano-images were obtained by the pseudoheterodyne interferometric detection: the scattered signal from the AFM tip was collected and demodulated at the third harmonics of the tapping frequency to obtain the genuine near-field response.

## Data availability
The data that support the findings of this study are available from the corresponding author upon request.

## Code availability
The codes that support the findings of this study are available from the corresponding author upon request.

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

## Acknowledgements

Work at Auburn University was supported by the National Science Foundation under Grant No. DMR-2238691, DMR-2005194, and ACS PRF fund 66229-DNI6. Support for hBN crystal growth was provided by the Office of Naval Research, award number N00014-20-1-2474. Q.M. and P.-J.H. acknowledge the support from AFOSR grant FA9550-21-1-0319.

## Author contributions

S.D. and M.C. conceived the idea and designed the experiments. M.C. performed the optical experiments. M.C., E.H., and Z.Z. prepared the samples. J.L. and J.E. provided the monoisotopic crystals. Y.Z., J.-S.W., X.L., and H.C. developed the theory. M.C. and S.D. analyzed the data. M.C. and S.D. wrote the manuscript with input and comments from all authors. S.D., X.L., P.J.-H., Q.M., and J.E. supervised the project.

## Competing interests

The authors declare no competing interests.
