## [Peer Review File · Nature Communications]

van der Waals isotope heterostructures for engineering
phonon polariton dispersionsREVIEWER COMMENTS

Reviewer #1 (Remarks to the Author):

The authors demonstrate engineering phonon polaritons via vdW isotope heterostructuring of hexagonal boron nitride (hBN). I like their idea, which is well supported by the excellent near-field imaging data. They indeed show a rather simple but efficient way to design the dispersion of hBN phonon polaritons. However, before I can suggest the acceptance for publication, I have two comments/suggestions on their work, see below:

1) Compared to the electric-gate- with graphene-hBN heterostructure (their previous work, ref. 46), I lack excitement on the impact of their results. Although polariton dispersion becomes more complicated in the isotope heterostructure, I do not see any remarkable advantage over other tuning method. It'd better to show such experiment result.

2) I like their near-field optical images shown Fig.2. However, I find that they are not easily to understand and interpret for a non-expertise reader. This is even worse for their Fig.3, I am easily lost in the complicated dispersion plots, with only noticing one message "complicated dispersion".

Reviewer #2 (Remarks to the Author):

In “van der Waals isotope heterostructuring showcased in engineered light-matter waves” Chen and coworkers, use scattering type near-field optical microscopy to study hyperbolic phonon polariton dispersions in heterostructures composed by h^{10} BN and h^{11} BN layers. Experimental and theoretical analysis is presented as a function of the number of layers in the heterostructure and as a function of their thickness.

The manuscript is generally well written and concise, while the data analysis is clearly explained.

s-SNOM has been used extensively to study phonon polariton dispersions in several 2D materials and their heterostructures. The effects of isotope enrichment in hBN of the polariton dispersion curves has also been demonstrated previously. A common criticism to polaritonic materials is that, in contrast to plasmons, their optical properties (e.g. dispersion relations) are in first approximation dictated by the composition of the material of choice. As the authors discuss, to a limited degree some tunability of these properties has been provided by substrate engineering and heterostructuring with other 2D and phase change materials.

The main novelty of this paper is the assembling of heterostructures from isotopically enriched layers (in this case: h^{10} BN and h^{11} BN) to obtain a “isotopically structured metamaterial” rather than isotopic homogeneous materials (naturally abundant or isotopically enriched). The work by Chen and al. is a great exercise in phonon polariton imaging and analysis (as many others in the literature) and indeed does achieve a more robust engineering of phonon polariton dispersions than previous attempts.

I think the idea is generally interesting but perhaps its application may benefit more other applications such as spintronics, since it is completely unclear how the modified phonon dispersions could be leveraged to substantially benefit optical applications. For example, the spectral range over which the polariton occur are modified only marginally by the isotopic engineering and heterostructuring. As analogy, engineering phonon polariton dispersions in 2D materials, in the literature and in this work, has been a sort of equivalent of starting a monochrome car and painting it with stripes of different colors, changing the headlights and wipers...; at the end of the day is pretty much the same car.

In summary, the work is robust, the main idea may be important in other applications but engineering phonon polariton dispersions to the degree demonstrated here has limited

prospects in practical applications.

Regardless of the editor decision for acceptance of the paper or otherwise, my (facultative) suggestion for the authors is to improve their title which is not very catchy. Perhaps something along the line: “isotopically hetero-structured metamaterials for engineering phonon polariton dispersions” or something like that.

Reviewer #3 (Remarks to the Author):

The manuscript „van der waals isotope heterostructuring showcased on engineered light-matter waves by M. Chen et al. reports on the measurement and calculation of phonon polariton waves and their dispersion relations in hexagonal boron Nitride (hBN) multilayers systems consisting of different isotopes. The authors exploit the existence of isotopically pure hBN crystals and flake to combine multilayers of ^{10}B and ^{11}B hBN with different thicknesses and stackings to create new layered “metamaterials” (although they don’t call it this way) with modified dispersion relations. They show multiple s-SNOM images of differently stacked isotope layer combinations and find a very good agreement between the experimentally determined and calculated dispersion relations for different isotope stackings and thicknesses.

The core idea of exploiting different isotopes when stacking van der Waals (vdW)-materials is new and very interesting and has, to my knowledge, not yet been exploited for dispersion engineering, although it is a straightforward extension of stacked vdW heterostructures. However, the manuscript is not well written and explained, e.g. it lacks a systematic explanation of the multiple dispersion branches and fails to give new insights into the underlying physics. Many graphs are crowded with information that is hard to understand for the reader as they are barely explained (e.g. Fig 2 with 4 s-SNOM images of many fringes and stackings, but no information where the polariton fringes are extracted for dispersion analysis). What shall the reader learn by showing 4 images and 10 barely explained, partially redundant dispersion relations? Important fundamental questions are not addressed, e.g.

- Can we extend the tuning range of hBN HPPs beyond the two individual dispersions of its constituent materials? What limits the accessible range?
- Can we understand the physics of the new branches, e.g. by symmetry arguments or by different signs of the permittivity tensors components in different frequency ranges?
- Are there new modes emerging at the interfaces?
- What happens for different rotation angles of the isotopically different hBN flakes?
- What are prospects and limitations of this methods?

Overall, the manuscript showcases an interesting idea, but fails to properly discuss and explain the data and the underlying physical mechanisms and restrictions. Thus, I cannot recommend it for publication in Nature Communications in its present form, a major revision is mandatory.

Response to referees' comments

Manuscript ID: NCOMMS-23-05894-T

Responses to Reviewer 1

We thank the reviewer for finding our results “*well supported by the excellent near-field imaging data*” and “*show a rather simple but efficient way to design the dispersion of hBN phonon polaritons*” and recommending the publication. In the following, we respond to the comments of this reviewer.

Comment 1-1: *Compared to the electric-gate- with graphene-hBN heterostructure (their previous work, Ref. 46), I lack excitement on the impact of their results. Although polariton dispersion becomes more complicated in the isotope heterostructure, I do not see any remarkable advantage over other tuning method. It'd better to show such experiment result.*

Response to 1-1: We thank the reviewer for this comment. The advantage of tuning hyperbolic phonon polaritons by isotope heterostructuring over Ref. 46 lies in unprecedented energy-momentum (ω - k) dispersions that break the universal response in all current hyperbolic systems. In current hyperbolic systems (including that in Ref. 46), the separations between adjacent hyperbolic polariton branches in the ω - k dispersion are identical ($\Delta k = \frac{i}{d} \sqrt{\frac{\epsilon_z}{\epsilon_t}} \pi$), and thus hyperbolic systems always possess the universal even-dispersion. Although Ref. 46 demonstrated the shift of polariton ω - k dispersion by coupling graphene and hBN, all hyperbolic branches are altered by the same degree and still follow the universal even-dispersion. In isotopic heterostructures reported here, hyperbolic polariton branches are separated unevenly and reveal a variety of unprecedented ω - k dispersions (Figure 3). These new ω - k dispersions can be further tailored by varying the composition, stacking, and thickness of building blocks in isotopic heterostructures to steer nanoscale light-matter wave propagation and to control the photonic density of states for nanoscale energy transfer.

The advantage of our work over the previous research on engineering hyperbolic polaritons has also been highlighted by Reviewers 2 and 3:

- “*This work is a great exercise in phonon polariton imaging and analysis (as many others in the literature) and indeed does achieve a more robust engineering of phonon polariton dispersions than previous attempts.*”
- “*The work is robust, the main idea may be important in other applications.*”
- “*The core idea of exploiting different isotopes when stacking van der Waals (vdW)-materials is new and very interesting and has, to my knowledge, not yet been exploited for dispersion engineering.*”

The discussion of unprecedented ω - k dispersions and advantages of isotopic heterostructures over current hyperbolic systems is detailed in two paragraphs at the end of Page 4 of the main text. To further emphasize the advantage of isotopic heterostructuring and its potential applications, we modified the perspective sentence in the last paragraph of the main text (Page 6). It reads: “*Therefore, the HPPs can possess new and tailorable energy-momentum dispersions and photonic density of states. These merits offer unique approaches to control nanoscale light-matter wave propagation, light emission, quantum optics, and energy transfer, thus expanding the current nanophotonic and thermal functionalities.*”

Comment 1-2: *I like their near-field optical images shown Fig. 2. However, I find that they are*

not easily to understand and interpret for a non-expertise reader. This is even worse for their Fig.3, I am easily lost in the complicated dispersion plots, with only noticing one message “complicated dispersion”.

Response to 1-2: We thank the reviewer for this comment. To address this comment, we have added the s-SNOM profiles (Fig. 2c-d) and corresponding descriptions to the main text. Additionally, we have added a paragraph to describe the new ω - k dispersions in ^{10}B - ^{11}B isotopic heterostructures (Fig. 3). These revisions are listed below:

a. On Page 3 of the main text, we have described the added s-SNOM profiles (Fig. 2c-d). It reads: *“These characteristics are revealed evidently in s-SNOM profiles (grey and green curves in Fig. 2c and 2d)—line cuts of the s-SNOM images (grey and blue dashed lines in Fig. 2a and 2b)—of the ^{10}B and ^{11}B hBN.”*

b. On Page 4 of the main text, we compare the short-period beats with long-period fringes. The corresponding text reads: *“As detailed in the s-SNOM profiles (Fig. 2c-d), the short-period beats in ^{10}B - ^{11}B isotopic heterostructures (black, red, blue, cyan, and pink curves) show narrower oscillation features than the fringes in single slab ^{10}B and ^{11}B hBN (grey and green curves).”*

c. On Page 4 of the main text, we have added a paragraph to describe the new ω - k dispersions in ^{10}B - ^{11}B isotopic heterostructures (Fig. 3). It reads: *“Remarkably, vdW isotope heterostructuring introduced here breaks this universal response by providing HPPs with a variety of new dispersions. In $\omega = 1350\text{--}1400\text{ cm}^{-1}$, $^{10}\text{B}|^{10}\text{B}|^{11}\text{B}$ (Fig. 3f) shows only 1 polariton branch instead of the conventional unlimited numbers of branches in hyperbolic materials. $^{10}\text{B}|^{11}\text{B}|^{11}\text{B}$ (Fig. 3c) and $^{10}\text{B}|^{11}\text{B}|^{10}\text{B}|^{11}\text{B}$ (Fig. 3j) show 2 branches with stronger low- k branches. $^{11}\text{B}|^{10}\text{B}|^{11}\text{B}$ (Fig. 3d) exhibits approaching branches: the 1st and 2nd (at the increasing k) branches are merging where the 1st branch fades at 1390 cm^{-1} while the 2nd one extends to the higher ω . $^{10}\text{B}|^{11}\text{B}|^{10}\text{B}|^{11}\text{B}$ in Fig. 4e and other compositions in the Supporting information S3 reveal merging branches: adjacent polariton branches merges. In addition, $^{11}\text{B}|^{10}\text{B}|^{11}\text{B}|^{10}\text{B}$ (Fig. 3i) supports unbalanced branches: the 2nd branch is significantly weaker than the other polariton branches.”*

d. In the caption of Fig. 2, we have described the added s-SNOM profiles. It reads: *“c–d, s-SNOM profiles cut from (a) and (b) reveal short-period beats in ^{10}B - ^{11}B isotopic heterostructures (black, red, blue, cyan, and pink) and long-period fringes in single slab ^{10}B and ^{11}B hBN (grey and green). L is the distance from the slab or heterostructure edge.”*

Responses to Reviewer 2

We thank the reviewer for finding our work *“the work is robust, the main idea may be important in other applications”* *“The main novelty of this paper is the assembling of heterostructures from isotopically enriched layers (in this case: h ^{10}B BN and h ^{11}B BN) to obtain a “isotopically structured metamaterial” rather than isotopic homogeneous materials (naturally abundant or isotopically enriched)”* *“The manuscript is generally well written and concise, while the data analysis is clearly explained”* and *“is a great exercise in phonon polariton imaging and analysis (as many others in the literature) and indeed does achieve a more robust engineering of phonon polariton dispersions than previous attempts”*. In the following, we respond to the comments of this reviewer.

Comment 2-1: *I think the idea is generally interesting, but perhaps its application may benefit more other applications, such as spintronics, since it is completely unclear how the modified phonon dispersions could be leveraged to substantially benefit optical applications. For example, the spectral range over which the polariton occur is modified only marginally by isotopic engineering and heterostructuring. In summary, the work is robust, the main idea may be*

important in other applications, but engineering phonon polariton dispersions to the degree demonstrated here has limited prospects in practical applications.

Response to 2-1: We thank the reviewer for this comment. Compared with all current hyperbolic systems with evenly separated ω - k dispersions, isotopic heterostructures show unevenly separated dispersions. They can be further tailored by controlling the composition, stacking, and thickness of the ^{10}B - ^{11}B building blocks. The advances of these unprecedented ω - k dispersions lie in the unique and tailorable nano-light propagation and photonic density of states. The latter can be utilized to control 1) nanoscale energy transfer for the dissipation engineering of electronic devices and 2) optical emission for novel light sources and quantum optics. To emphasize these potential applications, we modified the perspective sentence in the last paragraph of the main text (Page 6). It reads: “*Therefore, the HPPs can possess new and tailorable energy-momentum dispersions and photonic density of states. These merits offer unique approaches to control nanoscale light-matter wave propagation, light emission, quantum optics, and energy transfer, thus expanding the current nanophotonic and thermal functionalities.*”

In addition, the other fundamental parameter in isotopic heterostructuring—nuclear spin—is also very promising and can be explored to engineer material properties in many other fields, such as chemical reactions and spintronics mentioned by the reviewer. To incorporate the reviewer’s comment, we modified the main text’s last sentence and cited the related literature (Ref. 18 and 77–79) in spintronics. Now it reads: “*In addition, while this work mainly exploits spatially engineered atomic masses to showcase vdW isotope heterostructuring, the other fundamental virtue of this method—spatially engineered nuclear spins—is equally promising and can be explored in spintronics, chemical reactions, and quantum information and technologies.*”

Comment 2-2: *Regardless of the editor’s decision for acceptance of the paper or otherwise, my (facultative) suggestion for the authors is to improve their title, which is not very catchy. Perhaps something along the line: “isotopically hetero-structured metamaterials for engineering phonon polariton dispersions” or something like that.*

Response to 2-2: Following the reviewer’s suggestion, we have updated the manuscript title to “*van der Waals isotope heterostructures for engineering phonon polariton dispersions.*”

Responses to Reviewer 3

We thank the reviewer for finding our work “*The core idea of exploiting different isotopes when stacking van der Waals (vdW)-materials is new and very interesting and has, to my knowledge, not yet been exploited for dispersion engineering*” and “*the manuscript showcases an interesting idea*”. In the following, we respond to the comments of this reviewer.

Comment 3-1: *the manuscript is not well written and explained, e.g. it lacks a systematic explanation of the multiple dispersion branches and fails to give new insights into the underlying physics. Many graphs are crowded with information that is hard to understand for the reader as they are barely explained (e.g. Fig 2 with 4 s-SNOM images of many fringes and stackings, but no information where the polariton fringes are extracted for dispersion analysis).*

Response to 3-1: We thank the reviewer for this comment. We have responded to a similar comment in Response to 1-2. In addition, we have modified the paragraph on Page 5 of the main text. The corresponding insight text reads: “*As a result, the standard hyperbolic modes in ^{10}B and ^{11}B building blocks repulse and form new ω - k dispersions. Importantly, the delicate EM interactions and mode repulsion depend strongly on the composition, stacking, and thickness of*

the ^{10}B and ^{11}B building blocks, therefore offering the degrees of freedom to tailor HPPs in ^{10}B - ^{11}B isotopic heterostructures.”

Comment 3-2: *Can we extend the tuning range of hBN HPPs beyond the two individual dispersions of its constituent materials? What limits the accessible range?*

Response to 3-2: We thank the reviewer for raising this interesting question. In this work, the dispersion of HPPs in ^{10}B - ^{11}B isotopic heterostructures cannot exceed the Reststrahlen bands of h^{10}BN and h^{11}BN . Negativity permittivity in hBN is required to support HPPs, yet this is not possible out-side-of the Reststrahlen bands. To extend the accessible range, the N isotopes (^{14}N and ^{15}N) may be explored in addition to the B isotopes (^{10}B and ^{11}B) investigated in this work.

Comment 3-3: *Can we understand the physics of the new branches, e.g. by symmetry arguments or by different signs of the permittivity tensors components in different frequency ranges?*

Response to 3-3: As emphasized in our Response to 3-1, the emergence of new dispersion branches can be attributed to the repulsion of hyperbolic modes in h^{10}BN and h^{11}BN building blocks and their dependence as a function of ^{10}B and ^{11}B spatial separations. We have modified the corresponding text as detailed in Response to 3-1.

Comment 3-4: *Are there new modes emerging at the interfaces?*

Response to 3-4: We thank the reviewer for this question. As previous works (Ref. 21–26) demonstrated, phonon polaritons in hBN slabs are volume-confined modes with real momenta (non-evanescent propagation) along the vertical direction. In hBN isotopic heterostructures, the momenta are also real in the Reststrahlen bands. We did not observe interface-confined modes.

Comment 3-5: *What happens for different rotation angles of the isotopically different hBN flakes?*

Response to 3-5: We thank the reviewer for raising this excellent point. hBN is in-plane isotropic. Therefore the rotation angles of the h^{10}BN and h^{11}BN slabs won't affect the overall electromagnetic responses in ^{10}B - ^{11}B isotopic heterostructures. We note that moiré superlattices can form by rotating constituent single atomic layers in the heterostructures. In this work, the thicknesses of h^{10}BN and h^{11}BN vary from 5 to 35 nm, corresponding to the layer number of 15 to 100. Therefore, the effect of moiré superlattices is not evident. However, we do believe moiré superlattices are worthy of exploring in few-atomic-layer ^{10}B - ^{11}B isotopic heterostructures and comment in the last paragraph of the main text: *“It is also worth building few-atomic-layer isotopic heterostructures where isotopic moiré superlattices can be involved to configure related material properties, including moiré superconductivity, ferroelectricity, ferromagnetism, excitons, and many others.”*

Comment 3-6: *What are prospects and limitations of this methods?*

Response to 3-6: We thank the reviewer for this comment. The prospects of this method are mentioned in the last paragraph of the main text: *“Due to the universality and importance of isotopes, the method of vdW isotope heterostructuring established here can apply to a broad range of materials and engineer thermal, electronic, magnetic, quantum, and other properties where atomic mass or nuclear spin plays a role. Future works may be directed towards extending vdW isotope heterostructuring from hBN to other materials where a series of monoisotopic crystals have been produced. Yet, their isotopic heterostructures are promising but remain unexplored. It is also worth building few-atomic-layer isotopic heterostructures where isotopic moiré*

superlattices can be involved to configure related material properties, including moiré superconductivity, ferroelectricity, ferromagnetism, excitons, and many others. In addition, while this work mainly exploits spatially engineered atomic masses to showcase vdW isotope heterostructuring, the other fundamental virtue of this method—spatially engineered nuclear spins—is equally promising and can be explored in spintronics, chemical reactions, and quantum information and technologies.”

The main limitation of the method of isotopic heterostructuring, we believe, is the reliance on natural types of isotopes. Although there are various isotopes for each element, if a special atomic mass or nuclear spin of an element is needed, this may not be available from nature.

REVIEWERS' COMMENTS

Reviewer #1 (Remarks to the Author):

The authors improved the manuscript by adding the further descriptions. I suggest the acceptance of the manuscript.

Reviewer #2 (Remarks to the Author):

I find the manuscript by Chen et al., marginally improved upon revision. As before, I find the manuscript an interesting work conceptually. However, the authors did not make a strong case on how the modified phonon dispersions could substantially benefit optical applications in practice. Note that the modification of the phonon dispersions is qualitatively different (non-even spacing) than previous attempts (even spacing), but the spectral range is substantially the same. In summary, conceptually the work is novel, in practice the technical advance for optical applications seems marginally useful.

Reviewer #3 (Remarks to the Author):

The revised version of the Manuscript has indeed improved: The line profiles in the panel of the S-SNOM images and the more intense discussion of the modified polariton dispersions help the reader to better understand the topic.

However, I realized that reading the description of the complicated dispersion relations is still cumbersome and hard to understand, as many of the relevant basics are outsourced into the Supporting information. Although the authors answered most of my questions in their response letter, some of these questions are not properly addressed in the main text. and the manuscript could be improved. In addition, I have a few remarks, as readability and the scientific discussion of the manuscript currently can still be improved.

Overall, the manuscript can be accepted for Publication in Nature Communications after addressing of the minor comments below.

Comments:

1) The authors nicely explain the universal even-dispersion of isotope-pure hBN layers and the alteration of different polariton branches by the same Δk . Breaking this universal dispersion by isotope engineering is indeed the main novelty of this manuscript. However, a plot of this common dispersion is only shown in Fig S4, but not in the main text. It would be very helpful for the reader if this common dispersion is also plotted in the main text and directly compared with the new dispersions of the isotope heterostructures (including the limits of tuning, see, comment 3-2).

2) The discussion of the new branches in Fig. 3 is very hard to follow, as the authors refer to 1st and 2nd branches which are not labeled in Fig. 3. I tried multiple times to understand where the “approaching” branches are, but I failed. Which are the $N = -1, 0, +1$, etc branches in Figs 3? Again, the authors should try to give enough information in the main text so that a general reader can follow.

3) I was confused that the authors display some combinations like 10B|11B|11B in Fig. 3, where 2 flakes of the same isotope are listed separately. If the rotation angle of the individual hBN flakes does not matter (see my comment 3-5), then such a 3-layer stack should be equivalent to a 10B|11B (2-layer) stack with a different 11B thickness. An experimental comparison of such 2 layer stack with a 3 layer stack would proof the claim that in their thickness regime, Moire superlattices do not play a role.

4) In their response to comment 3-4, the authors say that they do not observe interface modes. However, in the calculated dispersion curves, at the interfaces of 10B and 11B flakes at the boundaries of the respective Reststrahlenbands, the permittivities of the surrounding medium changes sign, giving rise to the features at 1395 cm^{-1} (and presumably also at 1608 cm^{-1} , which is not shown). The role of this sign change is implicitly mentioned in the SI (see Figs. S5 and S6), but not properly mentioned or explained in the main text. However, from the dispersion graphs it is obvious that this sign change leads to many interesting features in the dispersion, with modes merging/ splitting / showing kinks. I suspect that there could be some interesting physics if one would take a closer look at the field distribution of these corresponding modes....

Response to reviewers' comments

Manuscript ID: NCOMMS-23-05894A

Responses to Reviewer 3

Comment 3-1: *The authors nicely explain the universal even-dispersion of isotope-pure hBN layers and the alteration of different polariton branches by the same Δk . Breaking this universal dispersion by isotope engineering is indeed the main novelty of this manuscript. However, a plot of this common dispersion is only shown in Fig S4, but not in the main text. It would be very helpful for the reader if this common dispersion is also plotted in the main text and directly compared with the new dispersions of the isotope heterostructures (including the limits of tuning, see, comment 3-2).*

Response to 3-1: We thank the reviewer for this helpful suggestion. Following the reviewer's advice, we have plotted the universal even-dispersion for homogenous hyperbolic systems in Fig. 4 and marked the evenly separated polariton branches with red arrows. We have also revised the corresponding text, which now reads: "Note that current hyperbolic systems have been following the universal even ω - k dispersion (Fig. 4): adjacent polariton branches are separated by identical momentum Δk (red arrows)."

Comment 3-2: *The discussion of the new branches in Fig. 3 is very hard to follow, as the authors refer to 1st and 2nd branches which are not labeled in Fig. 3. I tried multiple times to understand where the "approaching" branches are, but I failed. Which are the $N = -1, 0, +1$, etc branches in Figs 3? Again, the authors should try to give enough information in the main text so that a general reader can follow.*

Response to 3-2: We thank the reviewer for this comment. To better describe the unique polariton dispersions, we have marked the N numbers for polariton branches in Fig. 3–5 and used N to describe polariton branches in the ω - k dispersions. The revisions are listed below:

- a. On Page 4 of the main text, we have modified the description of N . It now reads: "..., and $N = 1, 0$, and -1 , etc., is an integer and varies at different dispersion branches (e.g., Fig. 3–4)."
- b. On Page 5 of the main text, we have modified the text on ω - k dispersions and used N to describe the polariton branches. It reads: "In $\omega = 1350$ – 1400 cm^{-1} , $^{10}\text{B}|^{10}\text{B}|^{11}\text{B}$ (Fig. 3f) shows only 1 polariton branch instead of the conventional unlimited numbers of branches in hyperbolic materials. $^{10}\text{B}|^{11}\text{B}|^{11}\text{B}$ (Fig. 3c) and $^{10}\text{B}|^{11}\text{B}|^{10}\text{B}|^{11}\text{B}$ (Fig. 3j) show 2 branches with stronger $N = 1$ branches. $^{11}\text{B}|^{10}\text{B}|^{11}\text{B}$ (Fig. 3d) exhibits approaching branches (see black arrows as a guide to the eye): the $N = 1$ and $N = 0$ branches are merging where the $N = 1$ branch fades at 1390 cm^{-1} while the $N = 0$ one extends to the higher ω . $^{10}\text{B}|^{11}\text{B}|^{10}\text{B}|^{11}\text{B}$ in Fig. 5e and other compositions in Supplementary Note 3 reveal merging branches: adjacent polariton branches merges. In addition, $^{11}\text{B}|^{10}\text{B}|^{11}\text{B}|^{10}\text{B}$ (Fig. 3i) supports unbalanced branches: the $N = 0$ branch is significantly weaker than the other polariton branches."
- c. On Page 5 of the main text, we have modified the text to describe the thickness-dependence of the ω - k dispersions. It reads: "The thickness of $27|23|35$ nm (for $^{11}\text{B}|^{10}\text{B}|^{11}\text{B}$, Fig. 5a) exhibits approaching branches: the $N = 1$ branch stops at $\omega \sim 1385$ cm^{-1} , whereas the $N = 0$ branch extends into $\omega > 1400$ cm^{-1} . The thickness of $25|27|16$ nm (Fig. 5b) exhibits

unbalanced branches: the $N = 1$ branch extends into $\omega > 1400 \text{ cm}^{-1}$, whereas the $N = 0$ branch is relatively weak and stops at $\omega \sim 1380 \text{ cm}^{-1}$.”

Comment 3-3: *I was confused that the authors display some combinations like $10\text{B}|11\text{B}|11\text{B}$ in Fig. 3, where 2 flakes of the same isotope are listed separately. If the rotation angle of the individual bHN flakes does not matter (see my comment 3-5), then such a 3-layer stack should be equivalent to a $10\text{B}|11\text{B}$ (2-layer) stack with a different 11B thickness. An experimental comparison of such 2 layer stack with a 3 layer stack would proof the claim that in their thickness regime, Moire superlattices do not play a role.*

Response to 3-3: The reviewer is 100% correct that a 3-slab $^{10}\text{B}|^{11}\text{B}|^{11}\text{B}$ can be equivalent to a 2-slab $^{10}\text{B}|^{11}\text{B}$ if the thicknesses of the h^{10}BN slabs and total thicknesses of the h^{11}BN slabs in two heterostructures are identical. In Fig. 3, we plot the $^{10}\text{B}|^{11}\text{B}|^{11}\text{B}$ (Fig. 3c) as one stacking case of the 3-slab heterostructure comprised of 1 h^{10}BN slab and 2 h^{11}BN slabs (Fig. 3c-e) in order to show that in addition to the composition (1 h^{10}BN slab + 2 h^{11}BN slabs), the slab stacking order is also an important parameter to engineer phonon polaritons, because the three stackings $^{10}\text{B}|^{11}\text{B}|^{11}\text{B}$ (Fig. 3c), $^{11}\text{B}|^{10}\text{B}|^{11}\text{B}$ (Fig. 3d), and $^{11}\text{B}|^{11}\text{B}|^{10}\text{B}$ (Fig. 3e) all exhibit unique ω - k dispersions. We thank the reviewer for mentioning the moiré effect again. Since the moiré effect is mainly at the interface between two atomic layers, we believe this effect is not evident for phonon polaritons in isotopic hetero-slabs where most composition slabs have > 30 atomic layers. However, we do believe moiré superlattices are worthy of exploring in few-atomic-layer ^{10}B - ^{11}B isotopic heterostructures and comment in the last paragraph of the main text: “*It is also worth building few-atomic-layer isotopic heterostructures where isotopic moiré superlattices can be involved to configure related material properties, including moiré superconductivity, ferroelectricity, ferromagnetism, excitons, and many others.*”

Comment 3-4: *In their response to comment 3-4, the authors say that they do not observe interface modes. However, in the calculated dispersion curves, at the interfaces of 10B and 11B flakes at the boundaries of the respective Reststrahlenbands, the permittivities of the surrounding medium changes sign, giving rise to the features at 1395 cm^{-1} (and presumably also at 1608 cm^{-1} , which is not shown). The role of this sign change is implicitly mentioned in the SI (see Figs. S5 and S6), but not properly mentioned or explained in the main text. However, from the dispersion graphs it is obvious that this sign change leads to many interesting features in the dispersion, with modes merging/ splitting / showing kinks. I suspect that there could be some interesting physics if one would take a closer look at the field distribution of these corresponding modes....*

Response to 3-4: We thank the reviewer for this insightful comment. We agree the permittivity sign change can lead to interesting phenomena at the heterostructure interfaces. A systematic study on this topic requires extensive numerical simulations on various interfaces and exploring an experimental method to extract the interface information from the overall heterostructure responses. We, therefore, believe this promising topic deserves an independent future work.

To highlight this promising topic raised by the reviewer, we have added a sentence in the last paragraph of the manuscript. It reads: “*In addition, it may be valuable to investigate the optical responses of interfaces within isotopic heterostructures, particularly at frequencies where the permittivity of the constituent slabs changes sign.*”